# An Exploration of Dance Learning Stress Sources of Elementary School Dance Class Students with Artistic Abilities: The Influences of Psychological Capital and Self-Concept

**DOI:** 10.3390/ijerph19127398

**Published:** 2022-06-16

**Authors:** Hsiu-Chin Huang, Huey-Hong Hsieh, Chia-Ming Chang, Yung-Chien Lu, Wui-Chiu Mui

**Affiliations:** 1Physical Education and Arts School, Chengyi University College, Jimei University, Xiamen 361023, China; op5166@yahoo.com.tw; 2Department of Leisure Management, Taiwan Shoufu University, Tainan 72153, Taiwan; nancylin809@gmail.com; 3Department of Physical Education, Health & Recreation, National Chiayi University, Chiayi 62103, Taiwan; gr5166@yahoo.com.tw; 4Department of Anesthesiology, Ditmanson Medical Foundation ChiaYi Christian Hospital, Chiayi 60002, Taiwan; s1080724@mail.ncyu.edu.tw

**Keywords:** art talent class, psychological capital, physical self-concept, stress of dance learning

## Abstract

The purpose of this study is to explore the factors which may cause the increase of students’ stress in dance class in elementary school. In this study, students’ demographic variables, psychological capital (which includes four sub-constructs), and self-concept (which includes five sub-constructs) were used as predicting variables to estimate their influences on dance class students’ stress level. A structured questionnaire was distributed to 450 elementary art talent class students with 412 valid responses. Structural equation modeling was used to test the relationships proposed by the study. As for demographic variables, the results show that the grade, gender, and the dance class hours per week had no significant influences on stress, while the seniority level had a negative influence, which indicated that junior dance students had more stress than senior students. As for psychological capital, self-efficacy and optimism had negative influences on stress, while the other two sub-constructs, hope and resilience, did not have a significant influence on stress. As for physical self-concept, the worry of overweight had positive influences on their stress, while appearance, physical ability performance, health status, and satisfaction of body parts had no significant influence on stress. Based on the research findings, suggestions were made to reduce students’ pressure in learning dance.

## 1. Introduction

In Taiwan, dance classes are offered in elementary, junior high, senior high schools, as well as in colleges and universities, which inspired and trained a large number of outstanding dance-related talents [1]. However, in recent years, the number of students enrolled in dance classes has been decreasing. One factor related to the decrease is the declining birthrate and the other is the learning stress which stopped many potential dancers from continuing [2]. Therefore, this study explored the factors associated with the dance learning stress among students in dance classes in order to provide practical suggestions for both students and teachers and prevent students from quitting.

In addition to the same academic subjects in regular classes, students in dance classes are also burdened with learning to dance. For example, dance presentations are held annually as a big show for which parents, teachers, and students have high expectations. To avoid letting them down, dancing students need to spend extra time in practicing [3]. As only the best performers can appear on the stage and play important roles, dance practice is demanding and highly competitive among peers. As a result, students suffer from physical strain and soreness due to constant movement repetition while maintaining their optimal physical performance during extensive dance practices [1]. For elementary school students, these stresses and burdens are not easy for them to cope with. How can schools, parents, and students themselves work together and cultivate a less-stressed environment so students can enjoy the classes and continue to go on? These are the main goals of this study. Therefore, the study aims to find related sources which may increase students’ stress in dance learning and practices and students, teachers, and parents can apply the findings and implications in this study to reduce their stress in their career pursuits. 

There are several factors that contribute to a decrease or increase personal stress. In this study, we focus on the factors that might directly affect elementary dance class students’ learning stress based on previous studies. Specifically, in this study, we were able to retrieve some related factors which include psychological capital, self-concept, and demographics.

Luthans, Youssef, and Avolio [4] defined psychological capital as a positive psychological state that individuals show as they grow and develop, and a resource that positively enhances the power of the mind. Therefore, this study was designed to determine whether psychological capital can reduce the stress of dance learning students in dance classes with artistic abilities in elementary schools.

Yeh [5] suggested that the physical self-concept is an individual’s perception of the capabilities of various elements of the body, including the perceptions of motor ability (motor skills, physical fitness, or physical ability) and physical appearance (tall or short, thin or fat, or personal attractiveness). However, in developing self-conception, children may feel stressed to learn when they feel the expectations of their parents and teachers, and competition among their peers.

Previous research [1,2,3] pointed out that students’ background variables may also have an impact on dance class students’ learning stress. Therefore, in this study, we employ students’ demographics as control variables and explored whether psychological capital and physical self-concept might influence their dance learning stress.

The study used a structured questionnaire to survey students’ perception related to upper mentioned constructs and used structural equation modeling to test the relationships among those variables which might contribute to students’ dance learning stress. The hypotheses proposed in the study were based on previous studies and are described in the following sections.

### 1.1. Relathionships among Students’ Demographics and Dance Learning Stress

As for elementary school students registered in dance classes, the study considered gender, grade, class, practice hours per week, and dance learning experience (years) as background variables (also known as control variables). The following will introduce previous studies related to those background variables and establish the corresponding hypotheses according to previous findings. First, we considered students’ gender. Gender stereotyping regarding dance often occurs in today’s society and culture [6]. Taking the dance classes of an elementary school in Chiayi City, Taiwan as an example, there are 140 students in the four classes of Grades 3, 4, 5, and 6, including 15 boys and 125 girls, which indicates that dance has been classified as a feminine sport, resulting in fewer boys engaging in dance activities. Lu [7] analyzed that among elite athlete students, girls were more distressed than boys in academic situations and were more likely to be stressed about further education than boys. Ko and Jheng [2] conducted a study on the stress of students from the dance classes of public senior high schools in Taiwan and found that female students felt more academic stress than male students. Therefore, this study made gender a control variable for the stress of dance learning. As for grade, Ko and Jheng [2] suggested that students in senior high school dance classes felt more stress than students in lower grades. Therefore, this study also considered grade a control variable, which may have some sort of impact on students’ dance learning stress. As for dance classes or practice hours per week, the Ministry of Education in Taiwan sets an average of 6 to 8 lessons of artistic talent specialization courses per week for students of Grades 3 to 6 [8]. However, many schools increase the number of practice times to more than 8 lessons per week; for example, in the case of presentations or summer and winter training. Therefore, the number of practice times per week was also included as a control variable in this study. As for dance experience, Ko and Jheng [2] also indicated that students in senior high dance classes with more than 10 years of dance experience were significantly less stressed than those with 5–9 years, 2–4 years, and less than one year of dance experience; the longer they danced, the heavier they felt they were, and the less stressed they were in maintaining their body functions to qualify them as good dancers. Therefore, the number of years of dance was included as a control variable.

In summary, we propose the following four hypotheses based on previous studies:

**H1-1.** 
*Gender affects students’ stress in dance.*


**H1-2.** 
*Grade affects students’ stress in dance.*


**H1-3.** 
*The number of practice hours per week affects students’ stress in dance.*


**H1-4.** 
*Years of experience in dance affect students’ stress in dance.*


### 1.2. The Relationship between Psychological Capital and the Stress of Dance Learning

In addition to their dance training, students also attend general classes, and they are often required to give performances representing their schools. Therefore, they often receive more attention and undertake more stress as compared with general students [2]. At a stage of basic skills learning, students in the dance classes of elementary schools have a lot of new dance movements to learn and rigorous dance tests to attend, which leads to their great psychological stress [1]. According to Chiang and Chen [9], psychological capital is a positive psychological ability and attitude that is built over an individual’s life development. Psychological capital was well-explored by Luthans, Luthans, and Luthans [10]; according to their studies, psychological capital comprises four aspects, namely, self-efficacy, hope, optimism, and resilience. In the following, we will present some previous studies of each aspect relating to dance learning stress and propose related hypotheses accordingly.

Self-efficacy: self-efficacy, also known as self-confidence, is the belief that an individual’s assessment of their ability to achieve a specific goal leads to their desired outcome [11]. Individuals with higher self-efficacy in sports are more willing and consistent in attempting to do more than they are capable of, thus, gaining higher achievements in sports [12]. Consequently, people with higher self-efficacy can improve their skills or perform regular tasks under pressure [13]. Therefore, the higher the self-efficacy of dance students, the lower the stress in dance learning. This study deduced H2-1, as follows: Self-efficacy negatively affects students’ stress in dance learning.

Hope: hope, which is an individual’s positive motivation, is a positive state based on the interaction between goals and paths [4]. Cho [14] noted that dance students in elementary schools were very concerned about outcomes that directly affect their perceptions of performance success and failure. For example, baseball is a sport with a high failure rate, and if players cannot immediately adjust their mindset to forget the previous failure and resume a positive offense and defense, they are not performing well [15]. Chen and Chi [16] studied 192 athletes in universities and found that the higher the perception of hope, the better the stress coping strategies of the athletes. Therefore, the higher hopes of dance students may lead to their lower stress in dance learning. This study deduced H2-2, as follows: Hope negatively affects students’ stress in dance learning.

Optimism: optimism is that individuals view events from a positive perspective or attribute and face their inner world with positive emotions, while negative events are attributed to external, unstable, and specific factors [4]. To attain certain success, athletes need to be trained and practice for years, and as Chang [17] pointed out, during their training, athletes are often faced with expectations, competition, injury, fatigue, and failure during their athletic careers. Optimistic people embrace stress positively and proactively, which helps to cushion them from personal stress. In the face of failure, they are less likely to become anxious and depressed; instead, they are motivated to persevere and train harder [18]. Following the logic, we can say the higher the optimism athletes withhold, the better their mindfulness, and they are capable of reducing personal errors under stress and sustain a high level of performance [13]. Therefore, for dance class students, we proposed the following hypothesis (H2-3): Students’ optimism is negatively related to dance learning stress.

Resilience: resilience is an individual’s psychological ability to bounce back from adversity and have the will to go beyond the original state during positive and challenging events [4]. Lin and Chen [19] investigated senior students in the physical education classes of elementary schools and found that students with more frustrating experiences were less tolerant of frustration because they were affected by failures. All good athletes possess the resilience to bounce back from failures; for example, Jones, Hanton, and Connaughton [20] studied 10 excellent international athletes (in swimming, track and field, gymnastics, middle-distance marathon, triathlon, and golf), and found that they had good mental resilience. Therefore, the higher resilience of dance students may lower their stress in dance learning. This study deduced H2-4, as follows: Resilience negatively affects students’ stress in dance learning.

### 1.3. The Relationship between Physical Self-Concept and the Stress of Dance Learning

Physical self-concept is an individual’s perceptual evaluation of the ability of body elements and an important predictor of motor engagement behavior [5]. According to Yeh [5], physical self-concept consisted of five constructs, namely, appearance, physical abilities, health status, worry of overweight, and satisfaction with body parts. The relationship between physical self-concept and stress in dance learning is described as follows and the related hypotheses are proposed accordingly.

Appearance: appearance refers to the importance students place on their appearance, clothing, and style. According to Yeh and Chang [21], a great body image is a goal for many young girls, and the demands for physical appearance are even higher for special groups, such as dancers, as they are often the center of attention and in the limelight, and their performances are often public displays that showcase their physical abilities, which makes the stage a unique environment to magnify their bodies. As a result, students in dance classes are more demanding of their physical appearance than other students of the same age [22]. Therefore, the higher appearance demands of dance students will increase their stress in dance learning. This study deduced H3-1, as follows: Appearance positively affects students’ stress in dance learning.

Physical abilities: Marsh and Peart [23] found that children’s perceptions of their physical abilities not only influenced their overall perceptions of self-efficacy, they also influenced later motor skills engagement and development. By definition, contrasting with mental ability, physical ability is the ability to perform some physical act which affects an individual’s engagement in motor skills, and those who are more physically capable are able to exercise for longer periods of time and are less likely to tire during strenuous exercises [24]. A dance performance is often seen as a competitive sport that requires a strong physical ability due to the use of movement and the influence of its execution [1]. Therefore, students with higher physical ability may have lower stress during the dance learning period. In this study, we used students’ self-concept of physical ability as measure of their actual physical ability, hence we proposed the following Hypothesis (H3-2): Students’ self-concept of physical ability had a negative influence on students’ dance learning stress.

Health status: health status refers to the dance students’ assessment of their physical health, such as whether they are often sick, whether they often feel muscle aches and pains, and whether they feel less healthy than their classmates. Dance students are often required to master a wide range of dance styles and techniques, which requires extensive training. Due to self-requirements and external factors (access to performance opportunities), students may suffer from acute and chronic injuries when they are fully committed to practice, and such injuries may result in students experiencing physical and psychological training difficulties, and cause the loss of performance opportunities [1,25]. Therefore, the poorer health status of dance students may increase their stress on dance learning. This study deduced H3-3, as follows: Health status negatively affects students’ stress in dance learning.

Worry of overweight: Athletes believe that weight loss is beneficial to athletic performance [26]. Past studies have found that athletes in many sports must constantly control their weight, such as female ballet dancers, ice skaters, and dance students who face weight control problems, which makes weight control a psychological stress for them [21,27,28]. Therefore, dance students’ increased worry of overweight may increase their stress in dance learning. This study deduced H3-4, as follows: The worry of overweight positively affects students’ stress in dance learning.

Satisfaction with body parts: Atalay and Gencoz [29] suggested that anxiety occurs when individuals are dissatisfied with their body because they are concerned about how others perceive their appearance; for example, when dancers do not gain recognition for their elegance, slenderness, and lightness, they may perceive that others have a negative impression of their appearance, and they will feel depressed [30]. Bane and McAuley [31] also pointed out that individuals who are dissatisfied with their bodies are more likely to have social body physique anxiety. Therefore, when dance students have higher satisfaction with their bodies, it may lower their stress in dance learning. This study deduced H3-5, as follows: Satisfaction with body parts negatively affects students’ stress in dance learning.

The five hypotheses, as derived from the previous theories, are shown in Figure 1.

## 2. Methods

In this section, we introduce the procedures performed to test our research hypotheses. First of all, we introduce the study subjects, namely, the selected participants from elementary school dance class students; second, we introduce the data collection tool, namely, a structured questionnaire developed based on research hypotheses; and third, we will introduce the statistical analyses performed using the data collected from the participants to test our hypotheses.

### 2.1. Participants

This study selected six elementary schools’ dance classes’ students with artistic abilities in Taiwan. Before investigation, all the administrators in charge of the six dance classes were well-informed of the purpose of this study and agreed to this survey. After agreements, the questionnaires were sent to schools to distribute to students. Students took the questionnaires home and got consent from parents then finished the survey. A total of 480 questionnaires were distributed and we were able to get 412 valid responses with a return rate of 85.8%. As for participants, only 31 (7.5%) were male and 381 were female (92.5%); 52 (12.6%) were third grade, 140 (34.0%) were fourth grade, 120 (29.1%) were fifth grade, and 100 (24.3%) were sixth grade. As for dance classes per week, 90 (21.8%) had six classes, 39 (9.5%) had seven classes, 195 (47.3%) had eight classes, and 88 (21.4%) had nine or more than nine classes. As for dance experience, 22 (5.3%) had 1 year of experience, 43 (10.4%) had 2 years of experience, 58 (14.1%) had 3 years of experience, 83 (20.1%) had 4 years of experience, and 206 (50%) had 5 years or more of experience.

### 2.2. Measurement

This study used a structured questionnaire based on research hypotheses for data collection. The questionnaire was comprised of four parts: the first part was demographical information, the second part was psychological capital scale, the third part was physical self-concept scale, and the fourth part was dance learning stress scale.

#### 2.2.1. Demographical Information

The demographical information included gender, grade year, dance classes per week, and years of dance experience which were all measured by categorical scales.

#### 2.2.2. Psychological Capital Scale

The psychological capital scale adopted the psychological capital scale for athletes developed by Chang and Chi [32], and the questions was rephrased to be in line with elementary schools’ dancing classes’ student scenarios. The scale was comprised of four constructs: self-efficacy (4 items), hope (4 items), optimism (4 items), and resilience (4 items). The scale was measured by a five-point Likert scale ranging from “strongly disagree = 1” to “strongly agree = 5”.

#### 2.2.3. Physical Self-Concept Scale

The physical self-concept scale adopted Wang and Wang’s [33] “The revision of the translated multidimensional body-self relations questionnaire” was rephrased for dance class scenarios. The scale was comprised of five constructs: appearance (6 items), physical ability performance (3 items), health status (5 items), the worry of overweigh (3 items), and satisfaction of body parts (9 items). The scale was measured by a five-point Likert scale ranging from “strongly disagree = 1” to “strongly agree = 5”.

#### 2.2.4. Dance Learning Stress Scale

The dance learning stress scale adopted Wang’s [34] primary students’ learning stress scale and was rephrased for dance class scenarios. The scale comprised of two constructs: dance and performance stress (6 items) and cram school dance learning stress (4 items). The scale was measured by a five-point Likert scale ranging from “strong disagree = 1” to “strongly agree = 5”.

## 3. Results

The study used structural equation modeling (SEM) to test the hypotheses. SEM is comprised of two models: measurement model and structure model. The measurement model basically reports the reliability and validity of the study instrument and the structural model reports the test results for hypotheses. The following sections report our results from the measurement model and the structure model, in that order.

### 3.1. Measurement Model

The reliability and validity of the study instrument were tested using WarpPLS 7.0 developed by Kock [35], which under PLS, provides two measures of item reliability: composite reliability and Cronbach’s. The convergent validity and discriminant validity were conducted to test validity of the instrument according to Hulland [36].

The factor loading of all items from PLS measurement model were all greater than 0.70 indicating good indicators. Composite reliability and Cronbach’s α values for all scales exceeded the minimum threshold level of 0.70 [37] indicating the reliability of all scales used in the study. As for convergent validity, the square root of average variation extract [37] of all values exceeded the minimum threshold level of 0.70 [37] indicating the reliability of all scales used in the study (Table 1). Fornell and Larcker’s test [37] for discriminant validity revealed relatively high variances extracted for each factor compared to the interscale correlations, which was an indicator of the discriminant validity of the nine constructs (Table 1).

### 3.2. Structure Model

According to Hulland [36], the structure model results should provide the analyses of the path coefficient test (hypotheses test) results and explanatory power. The following sections report the analyses of hypotheses test results and explanatory power.

### 3.3. Hypotheses Test Results

The evaluation of the structural model is used to examine the sixteen hypothesized relationships. The test results are shown in Figure 2. The test results are described in the following:

**H1-1.** 
*Students’ gender had no significant influence on dance learning stress (β1 = 0.01, p > 0.05).*


**H1-2.** 
*Students’ grade had no significant influence on dance learning stress (β2 = 0.07, p > 0.05).*


**H1-3.** 
*The number of students’ practice hours per week had no significant influence on dance learning stress (β3 = −0.02, p > 0.05).*


**H1-4.** 
*Students’ years of experience had a significant influence on dance learning stress (β4 = −0.14, p < 0.05).*


**H2-1.** 
*Self-efficacy had a significant influence on dance learning stress (β5 = −0.17, p < 0.05).*


**H2-2.** 
*Hope had no significant influence on dance learning stress (β6 = −0.03, p > 0.05).*


**H2-3.** 
*Optimism had a significant influence on dance learning stress (β7 = −0.17, p < 0.05).*


**H2-4.** 
*Resilience had no significant influence on dance learning stress (β8 = −0.07, p > 0.05).*


**H3-1.** 
*Appearance had no significant influence on dance learning stress (β9 = 0.03, p > 0.05).*


**H3-2.** 
*Physical ability performance had no significant influence on dance learning stress (β10 = −0.04, p > 0.05).*


**H3-3.** 
*Health status had no significant influence on dance learning stress (β11 = 0.01, p > 0.05).*


**H3-4.** 
*The worry of overweight had a significant influence on dance learning stress (β12 = 0.23, p < 0.05).*


**H3-5.** 
*Satisfaction of body parts had no significant influence on dance learning stress (β13 = −0.04, p > 0.05).*


### 3.4. Coefficient of Determination (R^2^)

R^2^ measures a model’s predictivity, which represents the explained variance and its influence on the structural model. The psychological capital, self-concept, and control variables all together showed an R^2^ = 0.24. It was suggested that R^2^ values must be above the threshold of 0.10 [38]. Therefore, the R^2^ values were above the threshold level of 10%, indicating a good predicting model as shown in Figure 2.

### 3.5. Discussion

This study found that the sources of elementary schools’ dance classes students’ stress was not affected by most of the control variables, such as gender, grade, and the number of classes per week, while the dance learning years had a significant negative path coefficient toward their dance learning stress, which indicates that students who have learned dance for fewer years have more stress than seniors. In Taiwan, if any elementary school wants to offer additional dance classes, they need to hire professional coaches/teachers and get approval from the Ministry of Education, Taiwan. After approval, they can recruit students willing to join the dance classes from Grade 3 or above from different classes or even from different schools. New students may need time to adapt to an unfamiliar environment. Since the seniors had more experience and might feel more comfortable in the environment, they might have had less stress than the juniors.

Regarding psychological capital, this study suggested that the self-efficacy and optimism of psychological capital negatively influence students dance learning stress, while hope and resilience’s path coefficients were not significant. As suggested by Luthans, Youssef, and Avolio [4], self-efficacy is an individual’s assessment of his or her ability to achieve a specific goal, as well as an expectation of his or her ability; if a student’s self-efficacy is inadequate, it leads to a belief that he or she will not be able to successfully overcome the stress of learning or achieve the expected outcomes. In elementary schools, most students learn to dance out of love, but have not yet decided if they want to become professional dancers; however, during performances, tests, or competitions, under the high expectations of peers, teachers, and parents, these students become stressed during their endeavors to be favored and fulfill expectations. When students fail to meet the expectations, they may feel frustrated and depressed, and such emotions may accumulate over time, leading to the loss of enthusiasm for dance, and even serious psychological disorders [1].

According to Chen, Lin, and Lin [39], self-efficacy is the self-confidence to put in enough effort to succeed when faced with challenges. Self-efficacy is based on practice and mastery, and when students repeatedly practice or are more skilled in a task, they tend to have higher self-confidence or self-efficacy; therefore, students in dance classes can enhance their self-efficacy through practice. On the other hand, teachers can also help students to increase their self-efficacy to reduce dance learning stress. For example, they can guide students to make daily schedules and encourage them to follow the schedules, and teachers can also encourage students to recognize the value of their hard work, which can increase their self-efficacy and self-confidence, thus reducing stress.

In addition, it is believed in Chinese culture that a strict teacher creates good students and that a teacher’s achievement comes from their students, thus, teachers are encouraged to be strict in education, and may severely criticize or scold students when their performances do not meet the expectation. On such occasions, optimism becomes important because it allows individuals to see a matter in a positive view or attribution and face their inner world positively, thus, lowering the stress in learning. Scheier and Carver [40] suggested that optimistic individuals have positive expectations regarding future matters and are willing to put more effort and persistence into the pursuit of their goals. Therefore, dance teachers can guide students to achieve self-confidence, and as a result of their hard work, students can gain a sense of superiority in the process of learning dance, and the joy of their hard work will be indispensable in their future lives, and they could choose dance as a career; thus, they become more motivated for their future development. According to Wang [41], there are three strategies for developing optimism. The first is to learn to reorganize and accept past failures, mistakes, and setbacks, to make positive attributions, and to develop an attitude of tolerance for the past. The second is to learn to appreciate the present, to be grateful for and content with the current training and competition performances, and to accept the current performance and status. The third is to adopt a positive, welcoming, and confident attitude towards future opportunities, and believe that hard work and training will definitely help achieve better results and development.

Dello and Stoykova [42] designed an intervention of psychological capital training students in Bulgaria, a one-month follow-up assessment of psychological capital training course to examine the durability of the training effects. The statistical analyses revealed significant improvements in the overall psychological capital after training as well as in each of its four dimensions, namely, self-efficacy, hope, resilience, and optimism. Therefore, it is suggested to add psychological capital training courses for students.

An examination of the physical self-concept found that appearance, physical ability performance, health status, and satisfaction with body did not have a significant influence on students’ dance learning stress. This is probably because students who attend dance classes in elementary schools share the same characteristics, such as the belief that appearance can be made beautiful by dressing up, and little difference was found among them. In elementary schools, students who are still growing in physique and physical ability performance hold that their physical ability performance will grow and their physique will improve as they grow up; thus, they may believe that their current satisfaction with their body and physical ability performance will not cause stress in learning.

This study found that the worry of being overweight according to the physical self-concept increased the stress in learning dance, as dance is an activity that uses the body as a language to convey specific ideas through a variety of movements in space, the performance of dance involves a lot of movement skills, and a lean physique is a key element in making the movements more skillful and graceful [43]. In addition, most audiences expect dancers to be light and graceful on stage, thus, dancers are constantly reminded to be light and graceful in order to fulfill the expectations and demands of society [44]. Therefore, as dancers usually practice in tight clothes and often look at their bodies in mirrors to correct their movements, for dancers, a slim body shape is like a “must”, that dominates their perceptions, thoughts, and feelings. Dantas et al. [25] and Liu [45] suggested that many female athletes seeking to maintain a slim body shape often employ erroneous and inappropriate eating behaviors, which may lead to malnutrition, amenorrhea, osteoporosis, and eating disorders [43,46]. In a study of the School of American Ballet students, it was found that 55% of its students failed to complete their four-year academic program and that poor eating habits were a major cause [47]. Therefore, it is important to instruct dance students in elementary schools to hold a proper sense of body shape, as well as proper eating habits and attitudes, so they can maintain healthy eating habits so they can stay in a healthy and good physical condition.

## 4. Conclusions

This study found that the variables of gender, grade, and the number of dance classes per week in elementary schools did not affect the stress in learning dance, while the years of dance learning negatively affected the stress in learning dance, meaning the stress is higher for children with fewer years of dance study. Teachers should therefore pay attention to the younger students in dance classes and provide them with appropriate support to reduce the stress of learning dance. It was found that, among the four aspects of motor psychological capital, self-efficacy and optimism have a negative influence on the stress in dance learning, and motor psychological capital is a psychological resource that can be trained and developed. Therefore, teachers can reinforce the self-efficacy and optimism of the students, which will reduce their stress in learning dance. Finally, it was found that only the worry of being overweight in the physical self-concept positively affected stress in dance learning, which means that teachers must help students learn how to control their weight to avoid the stress of being overweight during dance learning. It is recommended that a dietitian can be introduced to provide children with the correct diet and exercise concepts to control their weight in a healthy way, which will prevent them from adopting incorrect weight loss methods that may lead to ill health.

### Recommendation

According to the findings of this study, the recommendations for students of dance classes in elementary schools are that physiological capital can lower the stress to learn dance. Thus, it is suggested that students should improve their self-confidence and self-efficacy through repeated practice or by becoming proficient in dance techniques. It is also recommended that students should appreciate what they have learned, love what they have chosen, be determined in their goals, and build up their confidence in their commitment to the art of dance. In addition, they are encouraged to schedule their time, improve the efficiency of their studies and work, and recognize the value of their efforts to realize potential. In terms of optimism, students should learn to be tolerant of the past, appreciate the present and accept their current achievements and status, and remain positive about possible opportunities in the future. Finally, it is recommended that students learn to perceive their standard weight correctly, rather than just seeking a thin body shape, and that those who are truly overweight must learn to eat properly to control their weight.

The recommendations for teachers of dance classes in elementary schools are that dance instructors should work with academic subject teachers to understand students’ stress in dance learning and to care for students’ psychological capital and physical self-concept. By incorporating various aspects such as strengthening students’ self-efficacy and optimism regarding teaching and learning, students’ psychological capital can be enhanced, which is conducive to lowering their stress in dance learning and overcoming stress in life and academic studies. It is suggested that teachers offer psychological counseling to help students build up their self-confidence and maintain an optimistic attitude, and give them room to progress, which will gradually improve their self-efficacy. It is also recommended that teachers and parents understand whether the children worry about being overweight, and provide guidance on correct diets and exercise to control their weight through nutrition education. For children with fewer years of dance learning, in order to reduce the stress in learning dance, they should be given encouragement and have a modest attitude to become more confident.

As this study targeted students in the dance classes of elementary schools, we suggest that more complete studies could be conducted by collecting information on the stress of learning dance among students in junior and senior high schools. In addition, negative emotions, such as stage fright, the desire to win, and performance anxiety may also be important factors that influence dance learning stress during live performances with an audience, thus, future studies could incorporate positive and negative emotions to learn about the influences of emotions on stress in dance learning. Dancers are all expected to be absolutely correct and perform without flaws; therefore, further data could be collected to analyze whether the perception of perfectionism of dancers causes anxiety or stress on dancers exposed to such a concept in the long term.

## Figures and Tables

**Figure 1 ijerph-19-07398-f001:**
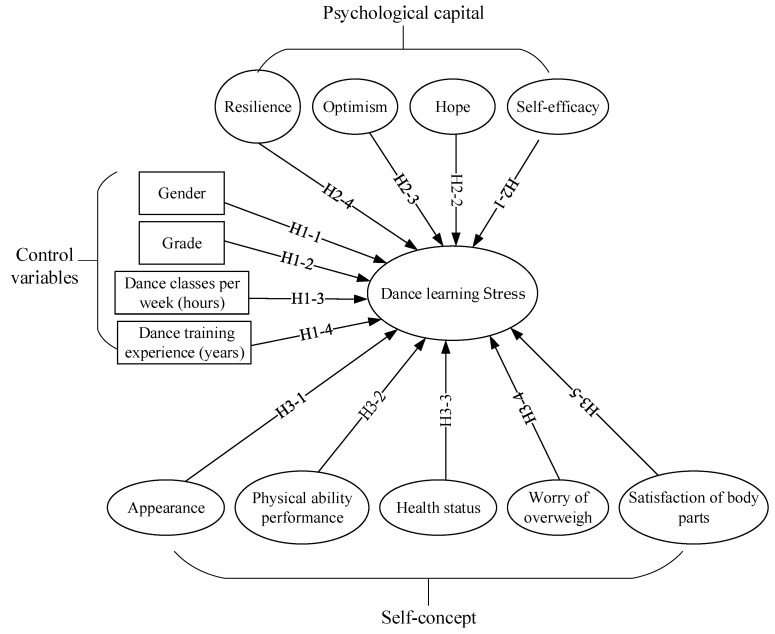
Framework of the research hypotheses.

**Figure 2 ijerph-19-07398-f002:**
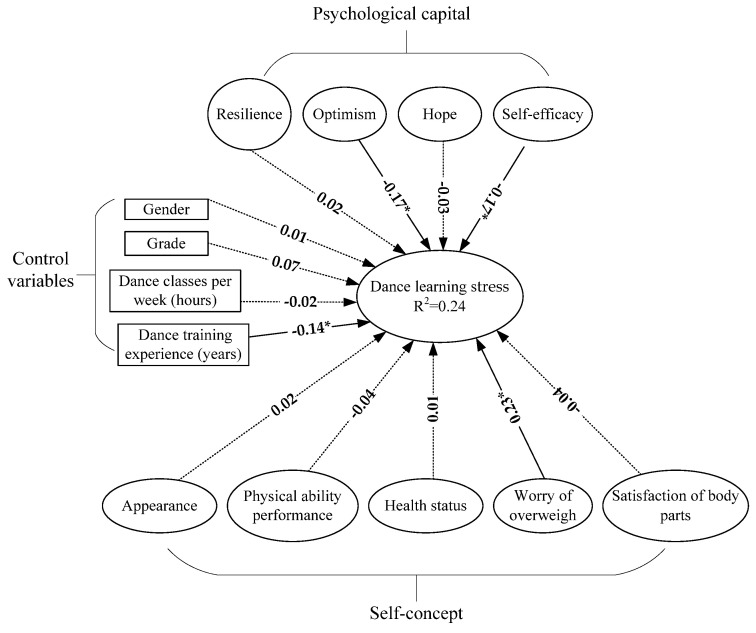
The path analysis. Note: The dotted line denotes that the tested standardized path coefficient was not significant; the solid line denotes that the tested standardized path coefficient was significant; “*” denotes *p* < 0.05; R^2^ is the coefficient of determination.

**Table 1 ijerph-19-07398-t001:** Reliability, convergent, and discriminant validity of measurement model.

Construct	(1)	(2)	(3)	CR ^b^	α ^c^
(1) PsyCap	0.61 ^a^			0.92	0.90
(2) SBC	0.45	0.63 ^a^		0.87	0.85
(3) DLS	0.43	0.33	0.65 ^a^	0.88	0.84

Note: PsyCap: psychological capital; SBC: self-body concept; DLS: dance learning stress; ^a^: square root of AVE (average variance extracted); ^b^: composite reliability; ^c^: Cronbach’s Alpha.

## Data Availability

Data can be provided upon request.

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
