# Peer review of "An Exploration of Dance Learning Stress Sources of Elementary School Dance Class Students with Artistic Abilities: The Influences of Psychological Capital and Self-Concept"

_ijerph, 2022, doi:10.3390/ijerph19127398_

Round 1

Reviewer 1 Report

Review Report

Article title: The Influence of Psychological Capital and Physical Self-Concept on Dance 2 Learning Stress of Elementary Schools’ Students in the Dance Classes with Artis- 3 tic Abilities

The authors aim to analyze the impacts of students’ psychological capital and physical self-concept upon the stress of dance learning from elementary art talent class in Taiwan.

This study explored the factors associated with the dance learning stress among students in dance classes so to provide practical suggestions for both students and teachers. The authors have made a significant effort in guiding the reader through the overall study.

Specific comments:

INTRODUCTION

I recommend adding studies from other parts of the world to the article as well. Before explaining the part of materials and method it is important to make a review of the scientific literature, following the scientific method.

METHODS

The research questions are missing.

The article does not list the specific questions of the questionnaire.

DISCUSSION

The discussion should be enriched with the results of foreign research in the light of scientific knowledge and compared with the results obtained by the authors of this article.

CONCLUSION and RECOMMENDATION 

Subchapter Recommendation should have the number 4.2, not 4.1.

Author Response

Point 1.

The authors aim to analyze the impacts of students’ psychological capital and physical self-concept upon the stress of dance learning from elementary art talent class in Taiwan.

This study explored the factors associated with the dance learning stress among students in dance classes so to provide practical suggestions for both students and teachers. The authors have made a significant effort in guiding the reader through the overall study.

Response 1.

Thanks for the comments.

Point 2.

Specific comments:

INTRODUCTION

I recommend adding studies from other parts of the world to the article as well. Before explaining the part of materials and method it is important to make a review of the scientific literature, following the scientific method.

Response 2.

Thanks for the comments. We had revised the article thoroughly and made it more comprehensible and logical

Point 3.

METHODS

The research questions are missing.

The article does not list the specific questions of the questionnaire.

Response 3.

Thank you for the comments. The questions were listed in section 2.2. We introduced our instrucments and questions there.

Point 4.

DISCUSSION

The discussion should be enriched with the results of foreign research in the light of scientific knowledge and compared with the results obtained by the authors of this article.

Response 4.

We revised the discussion carefully and made comparisons. Please read the discussion section. Thank you.

Point 5.

CONCLUSION and RECOMMENDATION 

Subchapter Recommendation should have the number 4.2, not 4.1.

Response 5.

Thank you. We had corrected it.

Reviewer 2 Report

Dear authors, I find your research very interesting, the hypotheses are clearly stated. Especially, I would like to point out a very clear figures. However, the text itself is sometimes hard to understand. In order to create a more "reader-friendly" text, I would recommend to give the article to an English speaker to make some corrections (besides spell check, (s)he could correct some complicated sentences).

I would also suggest to avoid the formulation "something affects students’ stress in dance learning", since the research does not show if e.g. hope affects stress or stress affects hope (which may, but also might not be the same). Instead, I would use something like "relation between hope and stress".

Author Response

Point 1.

Dear authors, I find your research very interesting, the hypotheses are clearly stated. Especially, I would like to point out a very clear figures. However, the text itself is sometimes hard to understand. In order to create a more "reader-friendly" text, I would recommend to give the article to an English speaker to make some corrections (besides spell check, (s)he could correct some complicated sentences).

Response 1.

Thanks for the comments. We had revised the article thoroughly.

Point 2.

I would also suggest to avoid the formulation "something affects students’ stress in dance learning", since the research does not show if e.g. hope affects stress or stress affects hope (which may, but also might not be the same). Instead, I would use something like "relation between hope and stress".

Response 2.

Thanks for the comments. Yes, we believe affect is a strong word. We use  “influence”, which indicates long term effects on something. In the study, the main purposes were to fine some sources relating to dance class student’s stress so we can make suggestions to students and teachers or parents to do something to reduce their stress so to reduce their dropping out intention and become good performers.

Reviewer 3 Report

While I applaud the authors on an interesting study and substantial sample size, there are several critical flaws which I believe preclude publication of this manuscript in its current form. Namely, some of the data presented in the results do not match the objective outlined in the introduction, the conclusions are not supported by the data, the overall tone of the paper appears to be suggesting causality despite its cross-sectional design, and explanation of findings are severely lacking references to support posited theories. Below I have included some specific comments:

Methods

  • Was gender assessed (i.e. self-reported identity) or was a binary category of sex reported?
  • Your hypotheses are all stated as if the data could provide conclusions of cause and effect (e.g. “grade ‘affects’ students’ stress” – the use of the word ‘affect’ implies a specific direction, when your data is cross-sectional so termporality cannot be ascertained)
  • Page 3, paragraph 2: are the performances in cheer squad comparable to those that exist in dance? Generally, dancers are expressing a variety of emotions that are relevant to their performance (not always a smile); and performance and learning are two different contexts (and I’d gathered so far that this paper is talking about the stress of learning, and not performing) so I am not sure the cheer squad comparison is entirely relevant here
  • Page 3, line 122: What is “Group A” in reference to athletes?
  • Page 3, line 134-135: This reads as a statement of fact but has no reference, so I assume it is a hypothesis?
  • Page 4, line 165-67: are you speaking about actual physical ability or perceived physical ability (2 separate constructs)? Seems to flit between these two ideas. Actual physical competence allows one to exercise for longer and are less likely to tire – do you have references for perceptions of physical ability being associated with the ability to exercise or longer without get tired?
  • Reading through the list of scales used in this study, I wondered what the reliability and validity was of these scales; then I was surprised to see some psychometric work presented in the results. It was not clear from the introduction that part of your objective was to evaluate these scales for use in this context. This should be made clear from the beginning for readers so they are not expecting this psychometric data for these specific scales to be presented in the description of the scales. However, some brief description of the validity and reliability of the original scales that were adapted should be included.

Results

  • this information about model building should be contained with in the methods section along with other details on the statistical analysis (i.e. sample size calculations, how missing data was handles, significance level, use of descriptive statistics, etc. )
  • Page 7, line 282: I believe this is a typo and that years of experience was significantly associated with dance learning stress
  • All language such as “influences”, “prediction”, “impact” implies a degree of causality which is not supported by a cross-sectional design
  • Section 3.4: what are you referring to by “behavioral intention and use behaviour”?; how are you defining “substantial” variance explained?

Discussion

  • Page 8, line 311-313: should clarify this is not what you hypothesized or what is apparently found in the literature according to your introduction (re: year of dance experience)
  • Line 316-318: do you have a reference to support this statement?
  • should discuss why hope and resilience were not associated with stress of learning dance
  • page 8, line 325-328 – do you have a reference for this statement? Also, link these points more specifically to self-efficacy and stress of learning dance
  • There is a lot of repetition of definitions throughout the discussion (repeated from the hypotheses section)
  • Pahe 9, line 339: I'm not sure I understand this reference to reducing "stress" in the general sense since your study appears o be focused on the stress of leaning dance (which presumably would not be remedied through leisure activities and reading).
  • Page 9, line 369-370: was there low variability in your sample among these variables that would support this thought?
  • Lines 371-374: include a reference for this statement
  • Section 4.1, line 404-408: are you implying that girls that struggle with their weight are the only ones who suffer from worrying being overweight? BMI or weight was not collected in this study, therefore you do not have data to support that girls that worry about their weight are the same girls that need to control their weight.

A major language edit is required to ensure proper grammar and reduce colloquial terms such as “doubles the stress” and “a slim body shape is like a spell”

Author Response

Dear reviewer:

   Thank you for the suggestions. We really appreciate your effort in helping us with the revision. In the following, we will provide responses to your comments respectively.   

Methods

  • Was gender assessed (i.e. self-reported identity) or was a binary category of sex reported?

Ans: We used a binary coding scheme for gender. “1” represents “male” and “2” represents “female”.

  • Your hypotheses are all stated as if the data could provide conclusions of cause and effect (e.g. “grade ‘affects’ students’ stress” – the use of the word ‘affect’ implies a specific direction, when your data is cross-sectional so temporality cannot be ascertained)

Ans: Well, it is a good question. Please allow us to explain how the assertions/ hypotheses were established. First of all, we appreciate your commenting word “affect”, so we change this term to “influence”, indicating all the predicting variables (in our cases, demographical variables, psychological capital and self-concept are the predicting variables) might have a long term effect on the outcome variables, which in our case, is the dance learning stress.

Second, it is common in the cross-sectional analysis to use the data collected to test derived hypotheses. For example, we may assert that Starbucks’ customers’ loyalty was affected/influenced by their brand image, service quality. Of course, the hypotheses were already supported by previous studies. So we may established that Starbucks’ customers’ perception of service quality may have a positive impacts on loyalty (H1) and also Starbucks’ customers’ perception of brand image may have a positive impacts on loyalty (H2). The following illustrate the hypotheses:

With the established hypotheses, all we need to do is to use a well-design questionnaire as our measurements to gather information related to customers’ perception during their stay at Starbucks. Then after we gather the information, we can use statistical tools to test/confirm our hypotheses and finally compare with previous studies and give suggestions to the brand owner for marketing strategies.

The similar procedures were adopted in our study.

  • Page 3, paragraph 2: are the performances in cheer squad comparable to those that exist in dance? Generally, dancers are expressing a variety of emotions that are relevant to their performance (not always a smile); and performance and learning are two different contexts (and I’d gathered so far that this paper is talking about the stress of learning, and not performing) so I am not sure the cheer squad comparison is entirely relevant here.

Ans: Thank you for the comment. That really point-out the flaw of our assertion. So we delete sentences related to squad cheering and revise this section as the following:

“Self-efficacy, also known as self-confidence, is the belief that an individual’s assessment of their ability to achieve a specific goal leads to their desired outcome [11]. Individuals with higher self-efficacy in sports are more willing and consistent in attempting to do more than they are capable of, thus, making higher achievements in sports [12]. Consequently, people with higher self-efficacy can improve their skills or perform regular tasks under pressure [13]. Therefore, the higher the self-efficacy of dance students, the lower the stress in dance learning.”

  • Page 3, line 122: What is “Group A” in reference to athletes?

Ans: Oops, this was a misplaced expression. We deleted it. Thank you.

  • Page 3, line 134-135: This reads as a statement of fact but has no reference, so I assume it is a hypothesis?

Ans: We had revised this section and made logical derivation of the hypothesis. Below are the revised article.

“Optimism is that individuals view events from a positive perspective or attribute and face their inner world with positive emotions, while negative events are attributed to external, unstable, and specific factors [4]. To attain certain success, athletes need to be trained and practice for years, and as Chang [17] pointed out, during their training, athletes are often faced with expectations, competition, injury, fatigue, and failure during their athletic careers. Optimistic people embrace stress positively and proactively, which helps to cushion them from personal stress. In the face of failure, they are less likely to become anxious and depressed; instead, they are motivated to persevere and train harder [18]. Following the logic, we can say the higher the optimism athletes withhold, the better their mindfulness and they are capable of reducing personal errors under stress and sustain a high level of performance [13]. Therefore, for dance class students, we proposed the following hypothesis (H2-3): Students’ optimism is negatively related to dance learning stress.”

  • Page 4, line 165-67: are you speaking about actual physical ability or perceived physical ability (2 separate constructs)? Seems to flit between these two ideas. Actual physical competence allows one to exercise for longer and are less likely to tire – do you have references for perceptions of physical ability being associated with the ability to exercise or longer without get tired?

Ans: According to the reference (Marsch and Peart), they point out students’ self-concept of physical ability is correlated to actual performance. So yes, in our article we are talking about the perceptions of physical ability not actual physical activity. To make it clearer, we had rephrased the sentences to avoid confusion. Thank you. Below are the revised version.

“Marsh and Peart [23] found that children’s perceptions of their physical abilities not only influenced their overall perceptions of self-efficacy, it also influenced later motor skills engagement and development. By definition, contrasting with mental ability, physical ability is the ability to perform some physical act which affects an individual’s engagement in motor skills, and those who are more physically capable are able to exercise for longer periods of time and are less likely to tire during strenuous exercises [24]. A dance performance is often seen as a competitive sport that requires a strong physical ability due to the use of movement and the impact of its execution [1]. Therefore, students with higher physical ability may have lower stress during dance learning period. In this study, we used students’ self-concept of physical ability as measure of their actual physical ability, hence we proposed the following hypothesis (H3-2): Students’ self-concept of physical ability had a negative influence on students’ dance learning stress.”

  • Reading through the list of scales used in this study, I wondered what the reliability and validity was of these scales; then I was surprised to see some psychometric work presented in the results. It was not clear from the introduction that part of your objective was to evaluate these scales for use in this context. This should be made clear from the beginning for readers so they are not expecting this psychometric data for these specific scales to be presented in the description of the scales. However, some brief description of the validity and reliability of the original scales that were adapted should be included.

Ans: Thank you for the comment. We revised this section. In the beginning of the method section, we started by introducing the procedures applied to test proposed hypotheses. Please refer to the beginning of the Methods section for details. As for the reliability and validity, the study performed structural equation modelling analyses to verify both. Please refer to section 3.1 for details. Thank you.

Results

  • this information about model building should be contained with in the methods section along with other details on the statistical analysis (i.e. sample size calculations, how missing data was handles, significance level, use of descriptive statistics, etc. )

Ans: Thank you for the comments. As pointed out in the article (section2.1, 480 questionnaires were distributed and we were able to retrieve 412 valid data. During the data acquisition period, we deleted responses with missing data. The reason for dropping these records was because we had high valid response rate (85.8%) and the statistical tools needed to test our hypotheses do not require large sample size (i.e. 100 samples can be applied for tests in performing PLS-SEM analyses.). The default significant level applied in social science is p=0.05 which is commonly recognized in the field. As for descriptive statistics, we had described the participants’ descriptive statistics in section 2.1 and since the purpose of this article is to test the hypotheses so we focused on the test results. Thank you. 

  • Page 7, line 282: I believe this is a typo and that years of experience wassignificantly associated with dance learning stress

Ans: Oops! Thank you for reminders. We corrected the typo accordingly.

  • All language such as “influences”, “prediction”, “impact” implies a degree of causality which is not supported by a cross-sectional design.

Ans: Thank you for the comments. As we explained in previous response. Please refer to “Method” section, Q2 and ans. Thank you. We replaced “influence” with “impact”. We also re-plot the figures to indicate which variables belong to control variables, psychological capital and self-concept. Thank you.

  • Section 3.4: what are you referring to by “behavioral intention and use behaviour”?; how are you defining “substantial” variance explained?

Ans: Thank you for the comments. We had revised this part as the following:

    “3.4. Coefficient of Determination (R2)

R2 measures a model’s predictivity, which represents the explained variance and its influence on the structural model. The psychological capital, self-consept and control variables all together showed an R2 = 0.24. It was suggested that R2 values must be above the threshold of 0.10 [38]. Therfore the R2 values were above the threshold level of 10%, indicating a good prediction as shown in Figure 2.”

Discussion

  • Page 8, line 311-313: should clarify this is not what you hypothesized or what is apparently found in the literature according to your introduction (re: year of dance experience).

Ans:  Thank you for comments. We had revised this parts accordingly. Please see the following:

“This study found that the sources of elementary schools’ dance classes students’ stress was not affected by most of the control variables, such as gender, grade, and the number of classes per week, while the dance learning years had a significant negative path coefficient toward their dance learning stress, which indicates that students who have learned dance for fewer years have more stress than seniors. In Taiwan, if any elementary school wants to offer additional dance classes, they need to hire professional coaches/ teachers and get approval from the Ministry of Education, Taiwan. After approval, they can recruit students willing to join the dance classes from Grade 3 or above from different classes or even from different schools. New students may need time to adapt to an unfamiliar environment. Since the seniors had more experience and might feel more comfortable in the environment, they might had less stress than juniors.”

  • Line 316-318: do you have a reference to support this statement?

Ans: Thank you for the comment. We had rephrased the sentences as followed from our findings and which make senses:

“New students may need time to adapt to an unfamiliar environment. Since the seniors had more experience and might feel more comfortable in the environment, they might had less stress than juniors.”

  • Page 9, line 339: I'm not sure I understand this reference to reducing "stress" in the general sense since your study appears to be focused on the stress of leaning dance (which presumably would not be remedied through leisure activities and reading).

Ans:  Thank you for the comments. We deleted the sentence and add suggestions for teachers. For example, to help students increase self-efficacy, teachers can “guide students to make daily schedules and encourage them to follow the schedules,  and also teachers can encourage students to recognize the value of their hard work, which can increase their self-efficacy and self-confidence so to reduce stress.”

  • Section 4.1, line 404-408: are you implying that girls that struggle with their weight are the only ones who suffer from worrying being overweight? BMI or weight was not collected in this study, therefore you do not have data to support that girls that worry about their weight are the same girls that need to control their weight.

Ans: Thank you for the comments. In our study, we collected information from both gender so in the paragraph, we did not mention only girls were concerned with their weight. In addition, we got the idea of students’ worry of overweight by asking if they are worry of overweight, which was to get an idea of how they think of their body image/self-concept not the reality. As we may know that in fashion zone, the fashion models are mostly too slim, but they still think they are overweight. That’s why in our study we suggest teachers to help students control their weights in a healthy way so they can reduce stress.

A major language edit is required to ensure proper grammar and reduce colloquial terms such as “doubles the stress” and “a slim body shape is like a spell”

Ans: Thanks for the comments. We had revised the article thoroughly.

Round 2

Reviewer 1 Report

Dear authors,

congratulations on improving your manuscript. You have significantly improved the clarity of your writing and have addressed most of my concerns.

 Kind regards.